# Assessment of the Current Status of Potyviruses in Watermelon and Pumpkin Crops in Spain: Epidemiological Impact of Cultivated Plants and Mixed Infections

**DOI:** 10.3390/plants10010138

**Published:** 2021-01-12

**Authors:** Celia De Moya-Ruiz, Pilar Rabadán, Miguel Juárez, Pedro Gómez

**Affiliations:** 1Centro de Edafología y Biología Aplicada del Segura (CEBAS)—CSIC, Departamento de Biología del Estrés y Patología Vegetal, P.O. Box 164, 30100 Murcia, Spain; cmoya@cebas.csic.es (C.D.M.-R.); mprabadan@cebas.csic.es (P.R.); 2Escuela Politécnica Superior de Orihuela, Universidad Miguel Hernández de Elche, Orihuela, 03312 Alicante, Spain; miguel.juarez@umh.es

**Keywords:** plant virus, mosaic viral diseases, Moroccan watermelon mosaic virus (MWMV), watermelon mosaic virus (WMV), cucurbits, viral disease prevalence

## Abstract

Viral infections on cucurbit plants cause substantial quality and yield losses on their crops. The diseased plants can often be infected by multiple viruses, and their epidemiology may depend, in addition to the agro-ecological management practices, on the combination of these viral infections. Watermelon mosaic virus (WMV) is one of the most prevalent viruses in cucurbit crops, and Moroccan watermelon mosaic virus (MWMV) emerged as a related species that threatens these crops. The occurrence of WMV and MWMV was monitored in a total of 196 apical-leaf samples of watermelon and pumpkin plants that displayed mosaic symptoms. The samples were collected from 49 fields in three major cucurbit-producing areas in Spain (Castilla La-Mancha, Alicante, and Murcia) for three consecutive (2018–2020) seasons. A molecular hybridization dot-blot method revealed that WMV was mainly (53%) found in both cultivated plants, with an unadvertised occurrence of MWMV. To determine the extent of cultivated plant species and mixed infections on viral dynamics, two infectious cDNA clones were constructed from a WMV isolate (MeWM7), and an MWMV isolate (ZuM10). Based on the full-length genomes, both isolates were grouped phylogenetically with the Emergent and European clades, respectively. Five-cucurbit plant species were infected steadily with either WMV or MWMV cDNA clones, showing variations on symptom expressions. Furthermore, the viral load varied depending on the plant species and infection type. In single infections, the WMV isolate showed a higher viral load than the MWMV isolate in melon and pumpkin, and MWMV only showed higher viral load than the WMV isolate in zucchini plants. However, in mixed infections, the viral load of the WMV isolate was greater than MWMV isolate in melon, watermelon and zucchini, whereas MWMV isolate was markedly reduced in zucchini. These results suggest that the impaired distribution of MWMV in cucurbit crops may be due to the cultivated plant species, in addition to the high prevalence of WMV.

## 1. Introduction

Cucurbits are among the most important horticultural vegetables in the Mediterranean basin. Melon (*Cucumis melo*), watermelon (*Citrullus lanatus*), cucumber (*Cucumis sativus*), zucchini (*Cucurbita pepo*), and pumpkin species (*Cucurbita moschata* and *C. maxima*) are all extensively and intensively grown throughout the area [1]. The sustainability of their crop production can be threatened by viral diseases [2]. Particularly, the occurrence of aphid-borne viruses appears to be expanding in Spain. This could be mainly due to the lack of effective countermeasures against plant viruses or their vectors, in addition to the increase in organic cultivation that may change the vulnerability of these cultivated plants to aphid-vectors. In this scenario, aphids are polyphagous pests that feed on several plant species that may affect the distribution and structure of viral populations infecting cultivated plants, with unpredictable epidemiological consequences [3,4]. Thus, there is a need to permanently examine the current status of viruses, as well as their genetic structure and evolutionary epidemiology [5] in order to cope with the often-emerging viral diseases. 

Mediterranean cucurbit plants are known to be infected by 28 viruses to date [2,6]. Among them, cucurbit aphid-borne yellows virus (CABYV; genus *Polerovirus*) is one of the most prevalent viruses [7,8,9,10]. In addition, potyviruses, such as watermelon mosaic virus (WMV), zucchini yellow mosaic virus (ZYMV), and papaya ringspot virus (PRSV), which all belong to the genus *Potyvirus*, as well as cucumber mosaic virus (CMV, genus *Cucumovirus*), among others, are the most commonly reported aphid-borne viruses affecting cucurbit crops [11,12,13,14,15]. WMV was initially described in Israel in 1963 [16], and subsequently, it has been reported in cucurbits crops from Italy, Tunisia, France [17,18], Spain [19], and Poland [20]. Currently, WMV is distributed throughout most of the European continent, and its transmission has been associated to at least 35 different aphids via a non-persistent manner, with the cowpea (*Aphis craccivora*)*,* cotton (*Aphis gossypii* Glover), and green peach (*Myzus persicae* Sulzer) aphids being among the most efficient vectors [21]. Based on the amino acid sequences of the CP protein, WMV isolates have been classified into three groups (G1 to G3) [13]. The G1 and G2 groups include isolates that have been referred to as the “classical” (CL) strain, which has been present in the Mediterranean basin for longest, with subsequent intraspecific recombinants between groups G1 and G2 [13,22,23]. The G3 group includes isolates that have been referred to as the “emergent” (EM) strain, and it has been associated with more severe symptoms observed since the year 2000 [13,14,23]. Further analysis has demonstrated that new genetic variants within the WMV groups can be found. These comprise CL (A and -B) and EM (1–4) subgroups [23]. In Spain, WMV is widespread in cucurbit crops and has become highly prevalent in melon and zucchini crops [12,19,24,25], with most of the isolates clustering into the EM group [26,27]. Aside from WMV, the occurrence of Moroccan watermelon mosaic virus (MWMV) has been determined in zucchini crops, and a relatively high importance have been reported in Spain, to such an extent that it is currently considered a re-emerging virus in the cultivation of cucurbits in the Mediterranean region [28]. MWMV was first identified as a strain of WMV affecting cucurbit crops in commercial-producing regions of Morocco, and was further considered a different species type or lineage from WMV [29]. Phylogenetic analysis showed three genetic groups (A-C): Clade A comprises isolates from Tunisia and Europe; clade B, isolates from South Africa; and clade C, isolates from Central Africa [30]. Similar to WMV, this virus is mainly transmitted by aphids, although tobacco aphid (*Myzus persicae nicotianae*) appear to be the most efficient vector followed by green peach aphid, and to a lesser extent cotton aphid and green citrus aphid (*Aphis spiraecola*) [31]. This re-emergent virus has been distributed throughout Africa, in addition to several regions of the Mediterranean area [32,33,34], and despite its long-term occurrence in Spain, its current status is unknown.

Different host plants and varieties along with different agro-ecological practices may alter the epidemiological patterns for viral diseases [35,36,37]. Therefore, the systematic monitoring of the viral diseases’ occurrence and their causative agents is essential. In addition, the understanding of the ecological epidemiology of the diseases can provide an approach for the early detection of the diseases and help to establish an efficient management and control system. Thus, the aim of this study was to increase the understanding about the presence and distribution of aphid-borne viruses that cause mosaic diseases in cucurbit species in Spain. We first monitored the occurrence of cucurbit-aphid borne viruses in watermelon and pumpkin crops. The monitoring was carried out for three consecutive seasons in three major producing-areas. Additionally, we used two WMV and MWMV isolates and constructed the cDNA infectious clones. We further infected five cucurbit plants with those clones, which allowed us to assess the symptom expression and RNA viral accumulation in order to understand the impact of the cultivated species and mixed infections on the WMV and MWMV distribution. 

## 2. Materials & Methods

### 2.1. Sample Surveys

A total 196 apical leaf samples were collected from symptomatic watermelon (*Citrullus lanatus*) and pumpkin (*Cucurbita moschata*) plants (Figure 1) grown in 49 different open-fields: 20 plots from Alicante (38°3′26″N, 0°51′17″O), 16 from Murcia (37°43′20″N 0°57′59″O), and 13 from Castilla La-Mancha (39°9′34″N 3°18′48″O) during field-inspections in the 2018, 2019, and 2020 seasons (Table 1). Field-inspections were carried out during July and August for each season. Four samples were collected from each plot and processed for total RNA extraction. All plant samples were stored frozen at −80 °C.

### 2.2. Cucurbit Virus Detection

Total RNA from the 196 plant samples was extracted using Tri-reagent (Sigma-Aldrich, St. Louis, MI, USA). Two replicates from each RNA sample were placed on positively-charged nylon membranes, and the RNA was fixed with an ultraviolet crosslinker. Dot-blot molecular hybridization was carried out using specific RNA probes for MWMV, WMV, CMV, PRSV, and ZYMV detection. Probes for WMV (CP), CMV, PRSV, and ZYMV were obtained previously in the lab [24]. One more probe corresponding to the P1 genomic region of WMV, in addition to other two probes corresponding to the CP and P1 genomic regions of MWMV, were synthesized for this study in order to double-check WMV and MWMV detection. DNA fragments from CP and P1 genes from each virus were amplified with primers: CE-2426 Fw 5′-ggcaacaattatgtttggag-3′ and CE-2427 Rv 5′-gtgttgaatatctcctatctccc-3′ (P1; WMV), and CE-2040 Fw 5′-tttttctctcactatgagttac-3′ and CE-2043 Rv 5′-tttttctctcactatgagttac-*3*′ (CP; MWMV), and CE-2424 Fw *5′*-ggctgcaattatgtttggtt-*3*′ and CE-2425 Rv 5′-gtaccatcttgtgcctcgc-3′ (P1; MWMV). Corresponding fragments were purified and subcloned into the vector pGEMT-easy vector following the manufacturer’s instructions to facilitate specific probe synthesis. Both specific probes were prepared by in vitro transcription incorporating digoxigenin [38]. The membranes were incubated overnight at 68 °C with the specific dig-labelled probes [8]. After the hybridization, a series of washes were carried out, followed by an incubation with the anti-digoxigenin antibody conjugated to phosphatase alkaline (Anti-Digoxigenin-AP, Roche Diagnostics, Germany) and the chemiluminescent substrate CDP-Star (GE Healthcare UK Ltd., England) [9]. The membrane analyses were performed using an Amersham™ Imager 600 chemiluminescent detector (GE Healthcare Bio-Sciences AB, Sweden). 

### 2.3. Full-Length MWMV and WMV Genome Amplification and Construction of Infectious Clones

MWMV- and WMV-infected cucurbit samples were randomly selected to carry out the construction of full-length MWMV and WMV infectious clones. Retrotranscriptions were performed using Reverse Transcriptase (Roche) with a specific primer either for MWMV CE-2043 5′-tttttctctcactatgagttac-3′ or WMV CE-2620 5′-tttttttttttttaggacaacaaacattaccg-3′. Then, the full-length WMV genome was amplified with PCRBIO VeriFi Mix Red (PCR BIOSYSTEM, UK) following the manufacturer’s recommendations, and the full-length MWMV genome was amplified with using Q5^®^ Hot Start High-Fidelity DNA Polymerase following the manufacturer’s recommendations [39,40]. The primers included an overlapping sequence from the pJL 89 vector. The MWMV primers were CE-2703 Fw 5′-*catttcatttggagagg*aaataaaacatctcaacacaac-3′ and CE-2714 Rv 5′-*atgccatgccgaccct*tttttttttttttttttctctcactatgagttac-3′, and the WMV primers were CE-2640 (5′-*catttcatttggagagg*aaattaaaacaactcataaag-3′ and CE-2641 5′- *atgccatgccgaccc*tttttttttttttaggacaacaaacattaccg-3′ (vector sequences are underlined). The pJL 89 vector was amplified with specific primers [41] using Phusion High-Fidelity DNA Polymerase (Thermo Scientific). The amplified ca. 9700 bp (MMWV-genome) and ca. 10,046 bp (WMV-genome), as well as ca. 4700 bp (pJL 89 vector) fragments were purified from a 0.7% agarose gel with the GENECLEAN^®^ kit (MP Biomedicals). MWMV and WMV genomes were cloned into the pJL 89 vector using a kit (In-Fusion^®^ HD Cloning, Takara Bio USA, Inc) following the manufacturer’s protocol [41,42]. Stellar™ competent *Escherichia coli* cells (Clontech Laboratories, Mountain View, CA, USA) were transformed, and each clone was named according to the reference sample; WMV-MeWM7 (Melon 2019) and MWMV-ZuM10 (Zucchini 2010).

### 2.4. Sequencing and Phylogenetic Relationships of MWMV and WMV Isolates

The full-length genomes of WMV and MWMV clones (MeWM7 and ZuM10) were sequenced by using internal primers and Sanger method (STAB VIDA, Caparica, Portugal). Their corresponding sequences were assembled using Geneious Prime 2020. Then, multiple sequence alignment was carried out using MUSCLE with the full genome sequences and 11 other complete genome sequences of WMV and MWMV, respectively, available in the NCBI/GenBank database. These aligned full-length nucleotide sequences were used to analyze the phylogenetic relationships among isolates by using the maximum likelihood method (MLM). The evolutionary distances were computed in MEGAX [43], using the maximum composite likelihood method with 1000 bootstrap replications and using the TIM2 + F+I + G4 model assigned by IQ-Tree based upon the Bayesian Information Criterion (BIC) minimal score [44]. The ML phylogenetic tree was inferred using ITol v5. The genetic distances are presented as the lengths of the branches, and only branches with bootstrap values >70% are shown. Additionally, the nucleotide and aminoacidic diversity of the MWMV-ZuM10 and WMV-MeWMV7 isolates were analyzed by comparing with the most recent full-length sequence from geographically close isolates: Sq10 1.1 (KY762266.1; M-WMV type) and Vera (MH469650.1; WMV type), using MEGA-X.

### 2.5. Agro-Inoculation of Cucurbit Plants Species

The *Agrobacterium tumefaciens* C58C1 strain was transformed with the purified plasmids that contained ZuM10 (MWMV) and MeWM7 (WMV) isolates. The cultures were incubated overnight at 28 °C, centrifuged, and resuspended into the same volume of MES buffer [45]. *Nicotiana benthamiana* plants were agro-inoculated with MWMV and WMV clones, and viral infection was verified after 21 dpai. Then, twelve healthy plants of each cucurbit species: Melon (var. piel de sapo), zucchini (var. black-beauty), cucumber (var. marketer), pumpkin (var. butternut), and watermelon (var. sugar-baby) were agro-inoculated with both MWMV-ZuM10 and WMV-MeWM7 clones separately and mixing (1:1), including three mock plants per each group of plant species. The inoculations were carried out in cotyledons of the cucurbit plant species, which were grown in a greenhouse (16 h photoperiod and 24 °C in a day/night cycle).

### 2.6. Viral Load Quantification

All systemic leaves of three plants (from those twelve inoculated) were individually harvested and grinded in a mortar using liquid N_2_ for each group of plant species and virus infection at each time-point (6-, 12-, 18-, and 24-days post-inoculation (dpai)). Total RNA was extracted using Tri-reagent, purified by phenol-chloroform extraction, and treated with DNaseI (Sigma-Aldrich, St. Louis, USA). The viral accumulation was estimated by measuring the viral RNA accumulation by absolute real-time quantitative PCR (RT-qPCR) with an AB7500 System (Applied Biosystems, Foster City, CA) using One-step NZYSpeedy RT-qPCR Green kit, ROX plus (NZYTech, Lisboa, Portugal). After a BLAST search and nucleotide sequence similarity analysis of WMV and MWMV isolates, the P1 gene sequence showed the lowest similarity between them. Then, the primers described previously for the synthesis of WMV and MWMV probes from the P1 gene were used to amplify the P1 fragment from both isolates, and those were cloned into pGEMt-easy vector in order to get viral RNA. These viral RNAs were used in a serial dilution (10-fold) to generate external standard curves for RT-qPCR. A new pair of primers were designed targeting MWMV P1 region (934–1099 nt); CE-2953 Fw 5′-caacgcgattgtgaaacg-3′ and CE-2954 Rv 5′-tccctggactcgaactg-3′, and also WMV P1 region (523–635 nt); CE-2959 Fw 5′-cacccaacctctgaaatgg-3′ and CE-2960 Rv 5′-ggctcagattttgccatc-3′ for RT-qPCR. Their specificity was monitored with melting curve analysis. The reaction mix was prepared according to the manufacturer’s instructions (NZYTech), and no-template controls were included to ensure product-specific amplification and the absence of primer-dimers. The RNA concentration in each sample (ng of viral RNA per 100 ng of total RNA) was estimated by plotting the threshold cycle (CT) values from each biological assay (n = 3, at each time-point) with three experimental replicates for each biological replicate.

### 2.7. Statistical Analyses

The analysis of the viral load for each plant species was performed with a General Linear Model (GLMs). Values from each plant were independent among treatments, but data were transformed with a logarithmic function to meet the assumption of normality and homoscedasticity of variance. We thus fitted the viral types (WMV and MWMV isolates), the five-cucurbit plant species, and the time of infection (dpai) as three-factor fixed effects, including replicates as random effects (REML) and using least-squares approximation. For the analysis of each WMV and MWMV type effects, virus accumulation from each plant species was considered separately and analyzed using a one-way repeated-measures ANOVA. Note that differences mean that the slopes of the regressions of viral load on plant species over time were different from each isolate and type of infection. All analyses were performed with the JMP software. *P*-value ≤ 0.05 is typically significant. Plot graphs of the viral RNA accumulation for each isolate and plant species were drawn using R software.

### 2.8. Nucleotide Sequence Accession Numbers

The complete nucleotide viral sequences of the MWMV (ZuM10) and WMV (MeWM7) isolates were deposited in GenBank under the accession numbers MW161172 and MW147356, respectively.

## 3. Results

### 3.1. Cucurbit Aphid-Borne Mosaic Viruses’ Occurrence in Watermelon and Pumpkin

WMV was the most common virus found in the plants showing mosaic symptoms from both crops (Figure 1 and Table 1). Overall, WMV was detected in 53% of the samples, M-WMV in 6%, PRSV in 4%, and ZYMV in 3%, while CMV was not detected. In watermelon, WMV was detected in 44% of the diseased samples from Castilla La-Mancha, 44% from Alicante, and 55% from Murcia. The presence of M-WMV was detected in 11% of the diseased samples from Castilla La-Mancha and 5% from Alicante, although it was not detected in the Murcia samples. No mixed infections were detected in watermelon samples. In pumpkin, WMV was detected in 37.5% of the diseased samples from Castilla La-Mancha, 63.6% from Alicante, and 66.6% from Murcia, while M-WMV was detected in 11.3% of the diseased plants from Alicante, 4.1% from Murcia, and not detected in plants from Castilla La-Mancha. CMV was not detected in these watermelon and pumpkin samples. Additionally, we detected mixed infection in pumpkin plants. WMV + PRSV was detected in 12.5% of the diseased plants from Castilla La-Mancha, while WMV + PRSV and WMV + MWMV were in 13.6% and 2% of the plants from Alicante, respectively, while WMV + ZYMV was detected in 25% of the plants from Murcia.

### 3.2. Genetic Analysis and Symptom Expression of the WMV and MWMV Clones

Melon (from Murcia 2019) and zucchini (from Murcia 2010) symptomatic leaves that tested positive for WMV and MWMV, respectively, were selected to amplify the full-length genomes. The amplified fragments were cloned, infectivity verified by using agro-inoculations in *N. benthamiana,* and sequenced. Both full-length genome sequences were aligned with other WMV and MWMV isolates from NCBI’s GenBank in order to examine the phylogenetic relationship between either WMV or MWMV isolates. For WMV, the phylogenetic tree showed that the WMV-MeWM7 isolate grouped together with EM group isolates (Figure 2A). The sequence analysis of the full-length genomes of WMV-MeWMV7 and WMV-Vera isolates (MH469650.1) from Spain indicated a 97% nucleotide similarity, and 86% between amino-acid sequences. Out of 221 nucleotide changes, 36 were non-synonymous mutations, with most of changes in the HC-Pro genomic region, in contrast to the 6K2 where changes were not observed (Appendix A). For MWMV, the phylogenetic tree showed that the MWMV-ZuM10 isolate grouped with the Tunisian and European isolates (group A), closely related to the Spanish Sq10 1.1 isolate (KY762266.1) (Figure 2B). Likewise, nucleotide sequence analysis of full-length genomes between MWMV-ZuM10 and Sq10 1.1 showed a 99% similarity, varying between 97–99% of similarity among each ORF. Out of 16 nucleotide changes, 8 were non-synonymous changes, with most changes found in the P1 genomic region (Appendix A).

To examine the symptom expression of both viral clones, a group of cultivated cucurbit plants (cucumber, melon, pumpkin, watermelon, and zucchini) were independently infected with either WMW-MeWM7 or MWMV-ZuM10 clones. Overall, after two weeks’ post-inoculation, all infected plants showed typical potyvirus symptoms; vein clearing and banding symptoms in leaves with the subsequent expression of mosaic patterns in older leaves, which rely on the virus/host (Figure 3). In particular, we observed that cucumber plants infected with the WMV-MeWM7 isolate showed a vein banding with mosaics (Figure 3F), while melon plants showed severe mosaics and also leaf mottling (Figure 3G). Infected pumpkins showed a mottling and vein clearing (Figure 3H), watermelon plants showed mild mosaics and leaf deformation (Figure 3I) and zucchini plants showed a mild mosaic and a sawn-edges on leaves (Figure 3J). In the case of the MWMV-ZuM10 isolate, we observed that infected cucumber, melon, pumpkin, and zucchini plants showed a severe vein clearing and banding with mild mosaics (Figure 3K,L,M,O), in addition to a sawn-edges on melon and zucchini leaves (Figure 3L,O), while the infected-watermelon plants only showed mild mosaics (Figure 3N).

### 3.3. Virus Accumulation of WMV-MeWM7 and MWMV-ZuM10 Clones in Single and Mixed Infections

We next sought to determine the viral load of both WMV and MWMV isolates in those cultivated plants and either in single or mixed infections in order to elucidate whether plant species and mixed infections have an influence on the occurrence of MWMV. The RNA accumulation of both MWMV and WMV isolates was estimated in five-plant cucurbit species by absolute RT-qPCR at 6, 12, 18, and 24 dpai. Both isolates infected all the cultivated plants species steadily. Overall, the viral load of both isolates increased over time, with significant differences between plant species and between single and mixed infections (Figure 4; F_4,177_ = 6.238 *p* < 0.001). In single infections, the average viral load of both isolates was dependent on the plant species (F_4,82_ = 18.361, *p* < 0.001). While the isolate WMV-MeMW7 accumulated significantly at higher levels in melon (F_1,13_ = 55.302, *p* < 0.001) and pumpkin (F_1,14_ = 23.477, *p* < 0.001), and marginally in cucumber (F_1,15_ = 5.025, *p* < 0.040), than isolate MWMV-ZuM10. The accumulation of MWMV-ZuM10 was only higher than WMV-MeMW7 in zucchini plants (F_1,17_ = 33.702, *p* < 0.001), and no differences were found in infected watermelon plants according to either isolate (F_1,15_ = 0.327, *p* = 0.575) (Figure 4A). Moreover, in mixed infections, we observed that the viral load of the WMV-MeMW7 isolate was greater than MWMV-ZuM10 in melon (F_1,16_ = 58.295, *p* < 0.001), watermelon (F_1,17_ = 12.313, *p* < 0.027), and zucchini (F_1,18_ = 77.669, *p* < 0.001). Whereas, the accumulation of MWMV-ZuM10 was only higher in cucumber (F_1,17_ = 5.369, *p* < 0.033), and both isolates accumulated at similar levels in pumpkin (F_1,17_ = 0.002, *p* < 0.965) (Figure 4B). In order to test differences on viral load under single and mixed infections, both isolates were analyzed separately for each plant species. The viral load of the WMV-MeMW7 isolate was significantly higher in cucumber (F_1,15_ = 23.484, *p* < 0.002) and zucchini (F_1,17_ = 5.21, *p* < 0.036) under mixed infections than single infections. Whereas, it was lower in pumpkin (F_1,15_ = 6.637, *p* < 0.021), and at similar levels in melon (F_1,15_ = 0.058, *p* < 0.81) and watermelon (F_1,15_ = 4.90, *p* < 0.05) (Figure 4). It is remarkable that the MWMV-ZuM10 accumulation in zucchini plants was significantly reduced in the presence of WMV-MeWM7 (F_1,18_ = 6.459 *p* < 0.020), while remaining at similar levels in cucumber (F_1,17_ = 0.003, *p* < 0.95), melon (F_1,14_ = 3.14, *p* < 0.09), pumpkin (F_1,16_ = 15.9, *p* < 0.05), and watermelon (F_1,16_ = 2.16, *p* < 0.15) under single and mixed infections. 

## 4. Discussion

Spain is one of the main producers of cucurbits in Europe. The production is concentrated in the central and southeastern areas, with around 63.000 ha under cultivation and a cucurbit production of 3.207.600 tm [46]. In our viral disease updates in three major producing areas, WMV was detected in 53% of the samples, MWMV in 6%, PRSV in 4%, and ZYMV in 3%, while CMV was not detected. These results suggest that WMV is the most prevalent aphid-borne virus causing mosaic diseases in these crops in southeastern Spain, as it also seems to be throughout the Mediterranean regions and Europe as a whole [14,20,47,48]. However, the high prevalence of WMV, in addition to the low occurrence of ZYMV, and even the non-presence of CMV, could be linked to our dates of sample collection. Whereas, environmental conditions may differ between those sampling areas, and hence, collection dates (July and August) may likely introducing a bias on the assessment of the status of viruses. However, sampling at the end of season is in line to maximize virus detection, and the occurrence of viruses was similar after three consecutive seasons. Nevertheless, some other reasons could also in part explain the results obtained in this study. For instance, it is likely that the use of potential commercial cultivars that are resistant/tolerant to potyviruses may be constraining these potyviral populations in squash crops [28,49]. Additionally, vector transmission rate and presence of wild plant species must be considered. On one hand, WMV is transmitted by at least 35 aphid species in a non-persistent manner [50,51] that could be favoring the occurrence of WMV. In fact, it has been shown that potyviruses can foster physiological and chemical plant changes, influencing vector-insect behavior [52]. It is likely that aphids preferentially settle on WMV-infected plants [53], and in turn, these host changes can be diverse for each cucurbit plant species, and can thus constrain the short- and long-distance dispersal of MWMV. It is interesting to note that MWMV was only detected in 2018 and no longer appears to be present. We speculate that the narrow host range of MWMV, and hence, the absence in alternative weeds might contribute to the short prevalence, in particular, during winter season [47,49,50]. It is therefore that either cultivated and wild plant species or virus transmission efficiency may have played an important role in shaping the population structure of potyviruses in these cucurbit crops.

Additionally, mixed infections of WMV + PRSV and WMV + ZYMV were detected in pumpkin plants, with negligible occurrence in watermelon plants. Given that multiple infections, including strains or different variants, are more common in nature than what would be expected to occur by chance [3,4,54], and their consequences on the epidemiology of plant diseases are unclear, it is difficult to infer the occurrence of mixed infections in cultivated plants, an in turn, to explain the absence of mixed infections in watermelon plants. We could speculate that the vulnerability of open-field crops to the viral aphid-vectors is rising in organic crop production systems, where chemical pesticides are banned. Thus, aphid vectors that are also polyphagous pests, feeding in a gregarious way on pumpkin plants, rather than watermelon, can have contributed to the occurrence of mixed viral infections. Nevertheless, traditional cucurbit cropping systems can have an ecological relevance to prompt mixed infections, as different cultivated plant species are often grown closely together, and this could favor viral dispersal within and between crops. It is therefore that mixed infections are gaining considerable significance in crop cultivations, and further research is required.

We next sought to determine the symptom expression and viral load of WMV and MWMV viruses through the construction of the two full-length infectious cDNA clones, WMV-MeWM7 and MWMV-ZuM10. Based on the full-length genomes, the WMV-MeWM7 isolate was grouped phylogenetically with the corresponding emergent clade (EM) [13,14,55], while the MWMV-ZuM10 isolate grouped within the clade A, which comprises isolates from Tunisia and Europe [32,33,34,56]. Both isolates showed a high similarity with closely related isolates from Spain; Vera (WMV type), and Sq10 1.1 (MWMV type), which ranged between 97% and 99%, respectively. Moreover, five cucurbit plant species, with relatively the same agricultural importance as cucurbit crops, were infected steadily with either WMV or MWMV cDNA clones, showing symptoms that differed between the viral species and host plants (Figure 3). Meanwhile, the WMV-MeWM7 showed severe mosaic and mottling symptoms in melon and watermelon leaves, the MWMV-ZuM10 isolate displayed mild mosaics in watermelon and severe vein banding with sawn-edges in melon, zucchini, and cucumber leaves. This symptom expression is consistent with symptoms described for other related WMV and MWMV isolates [28,57]. Thus, cultivated plants showing mosaics virus-like symptoms may be dependent on the cultivated species, in addition to the plant cultivar, growing conditions, or even the presence of other virus(es) infecting the same plant. In this sense, it should be noted that inspection of viral symptoms is an important issue, as the symptoms are often overlooked or misinterpreted as a nutritional deficiency. All of these aspects must be considered, as they could introduce a bias into the context of visual collection, and in turn, into the viral detection.

The analysis of the viral load revealed that plant species and infection type had a remarkable effect on the WMV and MWMV accumulation. WMV accumulated to higher titers than MWMV in melon, both single and mixed infections, as well as in watermelon and zucchini in mixed infections. Whereas, the accumulation of the MWMV type was to a lower extent than WMV, and showed only a higher viral load in zucchini plants in single infections (Figure 4). Thus, the WMV type appeared to be fitter than the MWMV type, and it is likely that this competitive advantage explains the prevalence of the WMV in cucurbit crops. Furthermore, the outcome of WMV and MWMV coinfections appeared to be variable, in particular for melon, watermelon, and zucchini plants. Whereas, the viral load of both viruses was similar in melon and watermelon plants (neutral interaction between each isolate), while it was reliably affected in mixed infections in zucchini plants. In particular, the viral load of WMV was increased by the presence of MWMV (synergistic interaction), while in contrast, MWMV was reduced by the presence of WMV (antagonistic interaction). Within-plant virus–virus interactions have been generally categorized as neutral, synergistic, or antagonistic, leading to unpredictable biological consequences that can have an impact on the disease outcomes and efficiency of control measures [4,58,59,60]. In this study, we observed neutral, synergistic, and antagonistic interactions of the same isolates in different host species, suggesting that virus–virus interactions can be contingent upon the ecological host. Thus, it is likely that the impaired distribution of MWMV observed in cucurbit crops during the last three seasons may be as a consequence of the increased cultivation of zucchini crops, compared to the pumpkin species, in addition to the wide distribution of WMV. In fact, the outcomes of WMV may occur in fields, not only influencing related viruses or strains, but it could also be altering the spread, distribution, and evolutionary dynamics of the viral populations of other viruses [57,61]. In this context, the combination of a potyvirus and a non-related virus appears to increase the accumulation of the non-related virus, as a consequence of the plant’s viral defense mechanism associated to HC-Pro [57,62]. In contrast, the combination of two potyviruses appears to be neutral on one side, while decreasing on the other side [63], which is in agreement with our particular results for zucchini plants, where WMV appeared to be steady and MWMV was antagonized in mixed infections. Nevertheless, it is worth mentioning that this analysis of virus accumulation in different plant species and mixed infections is not ruling out the possibility that differences in aphid transmission rates may have influenced the occurrence of MWMV in the field, as *Aphis gossypii* is a one of the most common aphids in cucurbits crops, and MWMV is greatly transmitted by *Myzus persicae* and to a lesser extent by *A. gossypii* [31]. The potential occurrence of aphid species would also need to be studied further.

In conclusion, we have observed that WMV is more prevalent than other potyvirus species in watermelon and squash crops in Spain, and that MWMV has an unadvertised occurrence. Despite the reach that the cultivated plant species can have on virus prevalence, our findings suggest that mixed infections between WMV and MWMV may have impaired the distribution of MWMV. Considering that cucurbit viral diseases are often spread through related plant species and vectored mainly by insects, either the composition of these cultivated plant species or mixed infections could play an important role in determining the prevalence and evolutionary dynamics of these viral diseases in cucurbit crops. This aspect demands further research on the occurrence, distribution, and genetic diversity of viral populations that affect different cultivated plant species, which is essential for the implementation of a comprehensive detection program and efficient control measures.

## Figures and Tables

**Figure 1 plants-10-00138-f001:**
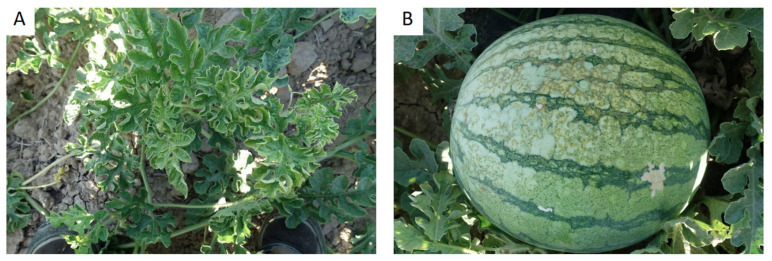
Mosaic symptomatic watermelon plants observed in the field (**A**), with a detailed picture of the affected fruit (**B**).

**Figure 2 plants-10-00138-f002:**
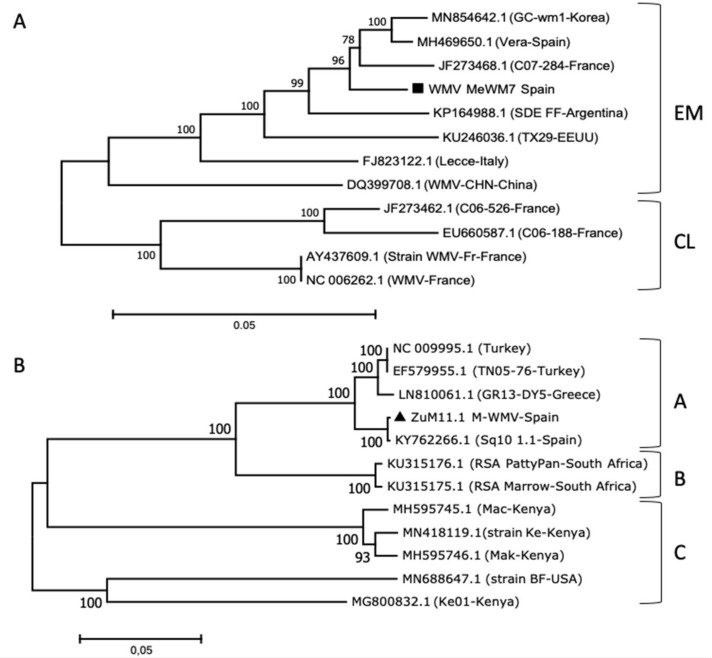
Phylogenetic relationships between complete genome sequences of the (**A**) WMV isolate MeWM7, and (**B**) MWMV isolate ZuM10, including complete sequences of another 11 full-length viral isolates available in the NCBI/GenBank database. The evolutionary history was inferred by using the maximum likelihood method, and the evolutionary distances were computed using the maximum composite likelihood method with 1000 bootstrap replications, using the General Time Reversible model. The Minimum Evolution tree was searched using the close-neighbor-interchange algorithm at a search level of 1. The genetic distances are presented as the lengths of the branches, and only branches with bootstrap values >70% are shown. Different clusters are indicated at the right for each corresponding virus.

**Figure 3 plants-10-00138-f003:**
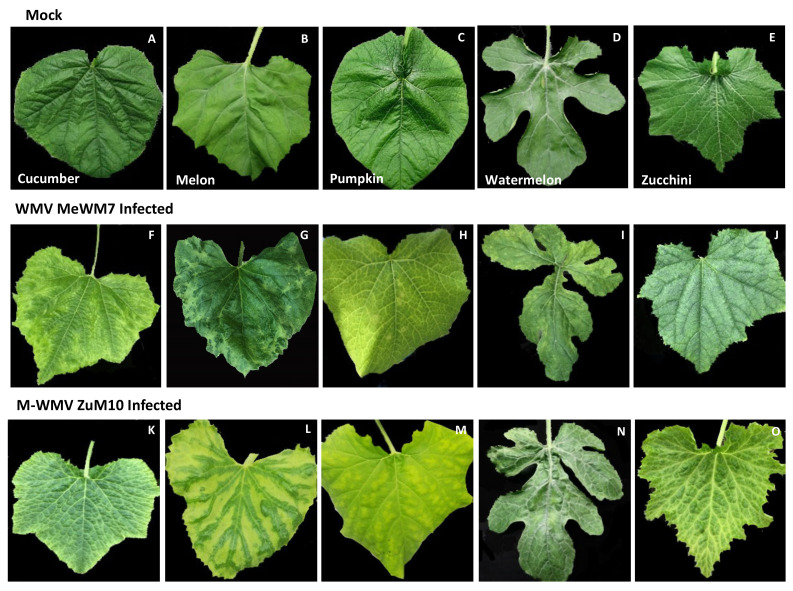
Symptom expressions of the agro-inoculated cucumber (**F**,**K**), melon (**G**,**L**), pumpkin (**H**,**M**), watermelon (**I**,**N**), and zucchini (**J**,**O**) plants with the WMV-MeWM7 and MWMV-ZuM10 clones, respectively. Mock plants (**A**–**E**). All plants were cultivated under experimental conditions in a greenhouse at 24 °C for 24 dpai.

**Figure 4 plants-10-00138-f004:**
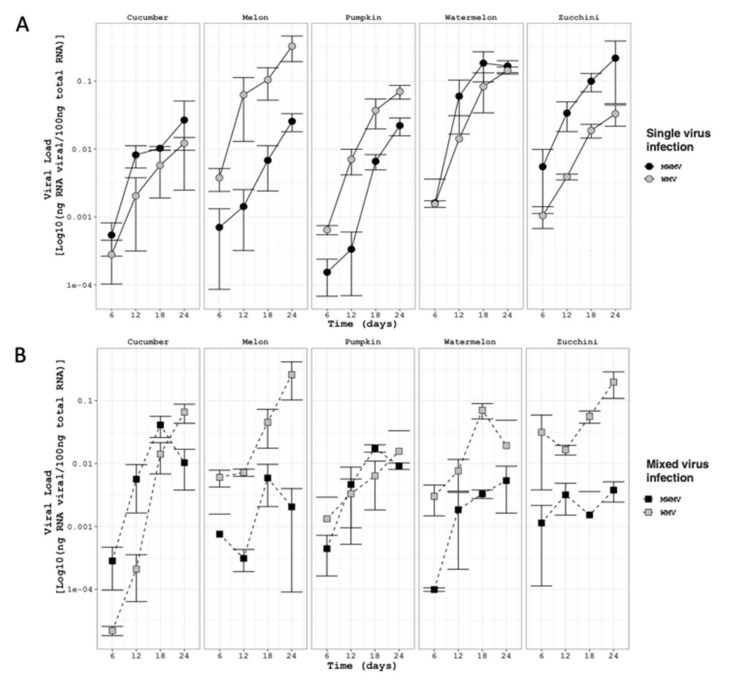
RNA viral load (Log10 (ng of viral RNA/100 ng of total RNA) mean and SE error bars, n = 3) of WMV-MeWM7 (grey) and MWMV-ZuM10 (black) in five different cucurbit plants under single (**A**, circles) and mixed infections (**B**, squares) during a time-course experiment. Viral accumulation of each clone was inferred by absolute quantification (RT-qPCR). RNA transcripts of the P1 gene were used in serial dilutions (10-fold) to generate external standard curves. RNA concentration was estimated from the threshold cycle (CT) values obtained from three independent plant samples, with three technical replicates for each sample, at each time-point (6, 12, 18, and 24 days post-agroinoculation).

**Table 1 plants-10-00138-t001:** Watermelon and pumpkin samples grouped by year and locality. Four symptomatic samples showing mosaics were collected from different plots, and two replicates per each plant sample were tested by dot-blot hybridization for watermelon mosaic virus (WMV), Moroccan watermelon mosaic virus (MWMV), papaya ringspot virus (PRSV), cucumber mosaic virus (CMV), and zucchini yellow mosaic virus (ZYMV). (-); non-detected.

					Positive Samples for Virus Tested				
Cucurbit Plant	Locality	Year	N° Plots	N° Samples Analyzed	WMV	M-WMV	PRSV	ZYMV	CMV
**Watermelon**	**Murcia**	**2018**	3	12	6	-	-	-	-
		**2019**	5	20	12	-	-	-	-
		**2020**	2	8	4	-	-	-	-
		TOTAL	10	40	22 (55%)	0 (0%)	0 (0%)	0 (0%)	0 (0%)
	**Alicante**	**2018**	4	16	10	2	-	-	-
		**2019**	2	8	-	-	-	-	-
		**2020**	3	12	6	-	-	-	-
		TOTAL	9	36	16 (44%)	2 (5%)	0 (0%)	0 (0%)	0 (0%)
	**C. La-Mancha**	**2018**	3	12	4	4	-	-	-
		**2019**	5	20	8	-	-	-	-
		**2020**	1	4	4	-	-	-	-
		TOTAL	9	36	16 (44%)	4 (11%)	0 (0%)	0 (0%)	0 (0%)
**Pumpkin**	**Murcia**	**2018**	2	8	-	1	-	-	-
		**2019**	1	4	4	-	-	2	-
		**2020**	3	12	12	-	-	4	-
		TOTAL	6	24	16 (66.6%)	1 (4.1%)	0 (0%)	6 (25%)	0 (0%)
	**Alicante**	**2018**	6	24	12	5	-	-	-
		**2019**	4	16	12	-	6	-	-
		**2020**	1	4	4	-	-	-	-
		TOTAL	11	44	28 (63.6%)	5 (11.3%)	6 (13.6%)	0 (0%)	0 (0%)
	**C. La-Mancha**	**2018**	1	4	4	-	2	-	-
		**2019**	3	12	2	-	-	-	-
		**2020**	0	0	-	-	-	-	-
		TOTAL	4	16	6 (37.5%)	0 (0%)	2 (12.5%)	0 (0%)	0 (0%)
**TOTAL**			49	196	104 (53%)	12 (6.1%)	8 (4%)	6 (3%)	0 (0%)

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
