# Peer review of "Assessment of the Current Status of Potyviruses in Watermelon and Pumpkin Crops in Spain: Epidemiological Impact of Cultivated Plants and Mixed Infections"

_plants, 2021, doi:10.3390/plants10010138_

Round 1

Reviewer 1 Report

The manuscript “Unadvertised occurrence of Moroccan-watermelon mosaic virus in watermelon and pumpkin crops in Spain: epidemiological impact of cultivated plants and mixed infections” intends to update the incidence of potyvirus (family Potyviridae) diseases affecting cucurbit crops in Spain. The study has been carried out in three important cucurbit-growing areas in Spain and includes two major cultivated species. Moreover, the sample collections have been performed in the last three seasons (2018-2020) giving a real picture of the current situation. The work also gives some indications from the epidemiological point of view, by means of biological assays and estimates of the titres of two viruses in single and mixed infections. For all these reasons, the manuscript properly fits the author purposes and contribute to point out the current status of potyvirus-disease epidemiology in the Mediterranean basin.

However, a main issue should be better addressed in the manuscript. The authors have focussed their results/discussion on the prevalence of WMV and the presence MWMV only, and they moved forward the other viruses, ZYMV, CMV and PRSV. These other viruses have been found in low numbers or they have not found at all. These low numbers sound to be an important result too, also considering that the previous surveys in Spain reported a different situation: for example, CMV has been a common virus in the past and now seems to be disappeared or ZYMV that showed low percentages and it is absent in watermelon but it is neglected in the result/discussion comments. Thus, the authors should discuss this part of their work thoroughly. For the same reason, they should provide more information about the sampling conditions such as seasonality, temperature etc… Temperature variations can affect the presence and incidence of the viruses: CMV and ZYMV can be more frequent during the early season whereas WMV and MWMV can be more frequent during the late season. Therefore the authors should indicate when the samplings have been performed during each year and if the three sampling areas are comparable for temperature trends. Another due comment is about the temporal variation of the sampling results through the three years; for example, MWMV has been found only in 2018, in both watermelon and pumpkin and in different areas. This is worthy to be commented. And what about the watermelon and pumpking varieties that were cultivated in the plots, did they include some resistant/tolerant varieties?

In my opinion also the title of the manuscript is a little bit misleading: can the only presence of low numbers of MWMV-infected samples be responsible for an epidemiological impact? I would choose a more inclusive title for this work.

The second issue is about the Discussion. In my opinion, the authors went into a long description of general epidemiological concepts: they speculated too much, and they did not deeply comment their own results. They should try to better stand out their findings against the general context.

Other comments/revisions are reported below.

INTRODUCTION

 Lines 47-54: the authors introduced many different elements that can contribute to the spread of aphid-borne viruses: organic cultivation, polyphagy of aphids, occurrence of mixed infections, emerging viruses… but the sentences do not sound consequential. Try to simplify this part, maybe retaining only the information that are relevant for the results obtained in this study. 

Line 48: increase IN organic cultivation.

Line 49: remove “viral”, and “...that feed ON…”

Line 55: the sentence sounds to be conflicting with line 45: 28 or 70 viruses? Anyway, the two sentences are redundant.

Line 55-60: I would suggest to introduce here some information about potyvirus symptoms, not only for an overview of the damages but also because the symptoms are important for the work since they have been used to select the samples during the field collection.

Line 57: “potyviruses” without upper case and report family and genus names within brackets.

Lines 64, 65, 86, 87: use the descriptor name for all the aphid species.

Lines 73-76: this sentence should be put in a temporal context, otherwise it is not clear why you hypothesize a displacement. Also, it is not clear how the second part of the sentence “whereas, EM transmission and accumulation has been reported to be higher than CL” is related to the first part.

Lines 76-80 and 93-95: the sentences “the occurrence of Moroccan watermelon mosaic virus (MWMV) has been determined in zucchini crops, and a relatively high importance have been reported in Spain, to such an extent that it is currently considered an emerging virus in the cultivation of cucurbits in the Mediterranean region [28].” and “MWMV was first described in 2009 in zucchini crops in southeastern of Spain, and despite its long-term occurrence, together with other cucurbit viruses, its current status is unknown.” could maybe be merged. But they raise a question: can MWMV be considered an emergent virus after more than 10 years from its appearance? Could the definition “re-emergent” be more appropriate?

Lines 88-93: this part is a little bit vague, try to go directly to the aim of your study.

Line 88: replace “heterogeneous genotypes populations” with something like “different host plants and varieties along with different agro-ecological factors…”.

Line 100: replace “cucurbit plants” with “cucurbit species”.            

MATERIALS AND METHODS

 Lines 106-110: I suggest: “A total 196 apical leaf samples were collected during field-inspections in 2018, 2019, and 2020 from symptomatic watermelon (Citrullus lanatus) and pumpkin (Cucurbita moschata) plants grown in different greenhouses and open-fields: 20 PLOTS from Alicante (38º3´26´´N, 0º51´17´´O), 16 from Murcia (37°43′20″N 0°57′59″O), and 13 from 108 Castilla La-Mancha (39°9′34″N 3°18′48″O) (Table 1).” As mentioned above, I suggest to add more information about the seasonality of the samplings since this can affect the virus incidence (i.e. CMV and ZYMV can be more present during the early season whereas WMV and MWMV more present during the late season). As well, a map of Spain localising the three areas could be useful, along with information about homogeneous/heterogenous environmental conditions of the three sites.

Legend of Fig 1: “two samples from each”, do you mean two replicates (= 2 dots) per each plant sample? It is not clear.

Lines 117 and 118: again, how many plant samples and how many RNA samples (replicates)?

Line 138: I think you should replace “symptomatic cucurbit samples” with “WMV- and MWMV-infected cucurbit samples…”.

Lines 170-172: you should clarify the choice of these two reference isolates: are they from Spain or are they the most recent full-length sequence available? Why are you analysing differences in their nucleotide and amminoaicd sequences: which is the means of this analysis?

Line 178: Twelve plants per each plant species (12 melons, 12 zucchini etc…)?

Line 187: How many plants per each plant species have been analysed for the viral load?

Line 196: replace “carried out” with “included”.

Lines 197-199: it seems that you checked the sequence similarity between the two virus after you performed the PCR… I think this part should be moved at the beginning of the paragraph, before the description of the target region and primer sequences.

Line 199-200: eliminate “corresponding” and “equivalent”.

RESULTS

Paragraph 3.1: replace mosaic diseased plants” with “symptomatic plants” or “plants showing mosaic symptoms”

Line 223: the WMV percentage seems to be 53 instead of 49%. Check carefully all the percentages reported in the paragraph.

Lines 223-224: what about ZYMV?

Line 224: remove “According to the cultivated plants”.

Paragraphs 3.2 and 3.3: when you refer to the clones, always report the virus they are from within the clone name, es: WMV-MeWMV7. In this way, it is easier for the reader to identify the clones through the paper (i.e  legend of fig 4 reports only WMV and MWMV and not the name of the clone whereas the text reports only the name of the clone).

Lines 243-244: are all this information relevant for the aim of the paper (i.e aa variability, syn ad non syn substitutions)? Genetic population analyses are not among the scopes, and the position of the two clones within each tree are enough to show their similarity to/distance from the other known isolates. At least, if you want to maintain this part, you must explain the choice of these two reference isolates.

Fig 2: both trees lack information about the genetic groups of the two viruses. For example, for WMV you must report which clades corresponds to emergent and classical groups otherwise the sentence in the text “the phylogenetic tree showed that the MeWM7 isolate grouped together with EM group isolates” cannot be understood. The same for MWMV.

Line 265: no upper case for potyvirus.

Lines 285-286: you do not have solid evidence of a displacement: too speculative (and anyway it is not correct to make speculations in the Result section: this is for Discussion).

Line 289: replace with “…between plant species and between WMV and MWVM…”

Line 292: looking at the figure 4, it seems that WMV titre is not higher than MWMV titre in cucumber. Please check. The same for lines 293-294.

Line 298: the viral loads in mixed infections in cucumber needs more comments. In general, the MWMV titre was higher than WMV titre until 18 dpai, but then the MWMV decreased. In my opinion you should provide ANOVA analyses for the 4 dpai separately and comment the new significant / no significant data.

Lines 300-303. Does this sentence refer to a comparison between single and mixed infection? It is not clear: re-write. Also, what about the other host species? Also no significant results should be reported!

Legend of fig 4: did you use CP or P1 gene for the standard curves?? And I think that the last sentence should be “RNA concentration in each sample was estimated from the threshold cycle (CT) values obtained from each independent biological assay, with three EXPERIMENTAL replicates at each time-point (6, 12, 18 and 24 dpai)”.

DISCUSSION

Line 316: again, what about ZYMV?!

Line 317: are you sure they are completely in agreement? You should consider putting some more thoughts on your field survey here.

Lines 320-325: try to better justify the prevalence of WMV: apart the high number of potential vectors, the other elements can be (weed role, absence of countermeasures etc…) would be valid also for the other potyviruses. Again, consider to better discuss the overall results of the field survey, not only for WMV and MWMV.

Line 329: replace “lack of occurrence” with “absence”.

Lines 326-347: this part is only speculative: it provides many different hypotheses that are poorly related among them and they are not supported by any data coming from this study. I suggest to move this part at the end of the discussion, primarily considering those hypothesis that can be supported by the results of your experiments.

Figure 1: the order of the figure in the manuscript is not correct.

Line 360: you are not comparing different viral isolates but different virus species. And replace “infected plants” with “host plants”.

Lines 360-363: the biological assay results are not discussed. How these results can be useful? How the symptomatology can be considered as a value added of this study? Better support your results!

Lines 364-365: you must support this hypothesis.

Lines 366-378: again, this part is very general and speculative. In this position, this part causes a hard break in the discussion of your results. Again, I suggest to move it at the end of the discussion (maybe you can marge this part with Lines 326-347), primarily considering those hypotheses that can be supported by your results.

Line 381: Why “however”?

Lines 381-383: “MWMV showed only a higher viral load in zucchini plants in single infections”. And what about cucumber?

Lines 383-384: “the WMV type was significantly fitter than MWMV type” Significantly for what? I think you cannot state a so general conclusion since your results are not significant for all the plant species. Be more precise.

Lines 385-389: it is not easy to interpret the trend of each virus in single and mixed infections, since in the results you did not provide any statistical analyses (except for zucchini) of the titre trend of WMV between single and mixed infection and the titre trend of MWMV between single and mixed infection. Try to analyse deeper the single vs mixed infection trends for each virus in Results, and then implement the discussion: only after that, you can talk about neutral/synergistic/antagonistic effects.

Lines 391-393: you stated that the cultivated cucurbit species LIKELY affect the MWMV and WMV distribution. Do you have any evidence of this? Do you know that the sampled plots were mainly planted with watermelon and zucchini rather than pumpkin, cucumber and melon, so that the incidence of WMV has been favoured?

Lines 395-405: again, too speculative. All these sentences are general and poorly supported by your results.                                                                                                        

Reviewer 2 Report

Mixed infections are currently one of the most serious problems in plant cultivation around the world. In the cultivation of cucurbits, Potyviruses appear to be the most serious threat. In this study authors have applied a global approach to investigate mixed infections in cucurbits, including crop monitoring in different regions of Spain, detection of the most important viruses in cucurbits, production of infectious copies of two viruses, and study of their accumulation in single and mixed infection. Overall, this study appears to be designed appropriately and all the outcomes have been discussed sufficiently. As this study has been planned and executed meticulously, I recommend only minor editorial corrections of the prepared manuscript.

line 81 'species' instead of 'specie'

line 252 between P1 genomic space is missing

in several places in the text there is a different font or a font of a different size (i.e. in the text covering lines 262-275)

Some headings have dots at the end, others don't.

Reviewer 3 Report

It was a pleasure for me reading this manuscript talking about mixed infections which has many ecological and evolutionary implications.

To my opinion, it would be better improving some parts of the paper.

Samples would Table 1 seems to be not enough exhaustive because it has the role to describe the materials and to show some initial data. The paper would gain in clarity if the authors could split these table into two because it is strenuous reading the first part of the results and referring to that table where data are cut to the bone and the total is not intuitive in order to check all percentages.

Line 119: the authors mentioned a Dot-blot hybridization but no results are shown, even that the dot-blot was carried out but not shown.

Line 162: my suggestion is to make a list (for example in a supplementary table) of all accession numbers used in the multiple alignment.

Line 184: it is not clear how the authors collected the samples, all systemic leaves were collected during the course of the viral infection. All systemic leaves in total after 6-, 12-, 18- and 24-days post-inoculation or all systemic leaves per plant? The authors are encouraged to make this sentence a little bit clearer.

Figure 2 requires much attention because of the poor quality and resolution. In the legend, line 259, the acronym ME is not specified.

Line 291: “the results showed that isolate MeMW7 accumulated significantly at higher levels in cucumber (F1,15=5.025, p < 0.040), melon (F1,13=55.302, p < 0.040), and pumpkin (F1,14=23.477, p < 0.001) than isolate ZuM10”: From the Figure it doesn’t seem like this. Would the authors correct it? The viral load is inverted in watermelon and zucchini, in the other species results have to be properly described and discussed.

The Standard Error in the Figure 4 is not clear to me: in some case the positive and the negative values are different.

I also strongly recommend to make a further table reporting F values to give this statistics the proper importance, clarifying in the Material and Methods the range and the meaning of this value.

I appreciated the dissertation on the virus-virus interactions.

Line 544: I would correct or change this reference, reporting the exact page if it is possible, because it is extremely complex to find those type of data in a general website.

The only weakness is recognized in total absence of any test or measurement on the aphids which have a significant role as many times repeated from the authors.

Reference 59 is already duplicated above.

The authors should also check the uniformity of the font size.

Round 2

Reviewer 1 Report

The manuscript “Unadvertised occurrence of Moroccan-watermelon mosaic virus in watermelon and pumpkin crops in Spain: epidemiological impact of cultivated plants and mixed infections” (now “Assessment of the current status of potyvirus in watermelon and pumpkin crops in Spain: epidemiological impact of cultivated plants and mixed infections”) intends to update the incidence of potyvirus (family Potyviridae) diseases affecting cucurbit crops in Spain. The authors have fully modified the manuscript according to the previous review and only few minor changes are needed:

Title: I would change potyvirus in potyviruses.

Line 57-58: the sentence “These observed changes can have unpredictable epidemiological consequences, affecting the distribution and structure of viral populations” can be now removed.

Lines 209-210: “The P1 gene fragment from both isolates was cloned into pGEMt-easy vector by using the primers described above.” I think in this new version of the manuscript, the primers are described after (below) this sentence.

Legend of fig 4: did you use CP or P1 gene for the standard curves?

Author Response

The authors have fully modified the manuscript according to the previous review and only few minor changes are needed:

Title: I would change potyvirus in potyviruses.

>Changed

Line 57-58: the sentence “These observed changes can have unpredictable epidemiological consequences, affecting the distribution and structure of viral populations” can be now removed.

>Removed

Lines 209-210: “The P1 gene fragment from both isolates was cloned into pGEMt-easy vector by using the primers described above.” I think in this new version of the manuscript, the primers are described after (below) this sentence.

>Clarified

Legend of fig 4: did you use CP or P1 gene for the standard curves?

>Changed. It was P1.

Many thanks, we really appreciate comments.